# Experimental Method for Simultaneous Determination of the Lamb Wave A_0_ Modes Group and Phase Velocities

**DOI:** 10.3390/ma15092976

**Published:** 2022-04-19

**Authors:** Olgirdas Tumšys

**Affiliations:** Ultrasound Research Institute, Kaunas University of Technology, LT-51423 Kaunas, Lithuania; olgirdas.tumsys@ktu.lt; Tel.: +370-37-351162

**Keywords:** ultrasonic lamb wave, A_0_ mode, group velocity, phase velocity, signal processing, dispersion curve segments

## Abstract

Determining Lamb wave dispersion curves when measuring phase and group velocity values at a fixed frequency is now a common and relevant task. In most cases, in order to solve such a problem, it is necessary to know the exact properties of the material, particularly its thickness. In experimental methods, Lamb wave parameters are evaluated directly from the test materials. This paper proposes a new and simple experimental algorithm for A_0_ mode group and phase velocity determination based on signal filtering and zero-crossing estimating. The main idea is to capture the zero-crossing instances of the signals closest to the signal envelope peaks and use these time instances to determine the phase and group velocities. The reliability of the proposed method was evaluated using simulated and experimental signals propagating in an aluminum plate. Theoretical modeling has shown that the proposed method enables the calculation of the A_0_ mode group and phase velocities with a mean relative error of less than 0.7%. An accuracy of 0.8% was observed during the experimental measurements.

## 1. Introduction

In recent years, researchers have extensively investigated Lamb waves for their potential use in structural health monitoring (SHM) and defect detection. These waves can be used to examine the entire thickness of thin-walled structures over large areas and to detect damage very sensitively [1,2]. The distinctive feature of these waves is their dispersive nature—the phase and group velocities vary with frequency. Therefore, when measuring phase and group velocities, questions arise not only about the value of the velocity itself, but also about the corresponding frequency. In the general case, these dependencies are defined by the dispersion curves. Therefore, the determination of dispersion curves and the measurement of the values of phase and group velocities at a fixed frequency is currently a very common task, the solution of which is significant for the application of Lamb waves.

In general, Lamb waves have an infinite number of symmetric and antisymmetric modes that differ in their propagation properties. However, in most cases, to simplify signal processing, only the fundamental modes (A_0_ or S_0_) are exploited. The advantage of both fundamental modes is that they propagate in a wide frequency range, their excitation is simple, and the peculiarities of their propagation have been widely studied in the literature [3,4,5]. At the same time, research is continuing to develop new methods and methodologies for measuring the phase and group velocities of fundamental modes. These methodologies usually focus on either group velocity or phase velocity estimations. Analytical, numerical and experimental methods are used to determine these velocities. However, analytical and numerical methods require knowledge of the properties of the materials in which Lamb waves propagate. Meanwhile, in experimental methods, Lamb wave parameters are evaluated directly on the basis of the test materials.

In experimental methods, the group velocity of Lamb wave propagation is widely measured using the Hilbert transform to determine the difference in time-of-flight (ToF) between the two positions [6]. However, this method makes it difficult to measure ToF for overlapping modes and reflected signals. Therefore, a variety of new experimental methods for estimating group velocity have emerged. One such method is the short-time Fourier transform (STFT), which can be used to estimate the group velocity of the Lamb wave [7,8]. STFT is used to extract information from the signal in the time-frequency domain in order to measure the arrival time of single-frequency components. Based on the results of these frequency measurements, the dispersion curves of the group velocity are reproduced. However, the resolution of the STFT is controlled by the window length and this is limited [7]. For the same purposes, many authors have used wavelet transform (WT) and Wigner–Ville distribution (WVD) [8].

Two alternative approaches (two-dimensional fast Fourier transform (2D-FFT) and methods of time-frequency processing) were used in [9] to determine the group velocities of zero-order Lamb wave modes in aluminum plate. 2D-FFT provides approximated dispersion curves in the k-f domain, which correlate with dispersion curves calculated via the solution of the Rayleigh–Lamb equation. The authors in [10] presented a new approach for the direct calculation of group velocity curves using the wave and finite element (WFE) scheme. Their model is computationally efficient and can be used for plate-like structures of arbitrary complexity. In [11], two kinds of time–frequency domain methods for broadband Lamb waves were proposed. The first one is based on the concept of the general parameterized time–frequency transform (GPTFT). The other one is called the time–frequency de-dispersion transform (TFDT). The proposed method shows great robustness in regard to inaccuracies in the dispersion data.

It should be noted that STFT, WT, WVD and the other methods mentioned above provide information on group velocity dispersion curves.

Many methods that are more varied have been proposed for measuring the phase velocity of Lamb wave propagation. These include frequency-wavenumber domain filtering [12,13,14], the matrix pencil method [15,16], the non-contact hybrid method [17,18], the zero-crossing method [19,20], etc. A similar method, using the frequency and wavenumber mapping of the dispersion curves, is called multi-signal classification (MUSIC) [21]. Two-dimensional fast Fourier transform (2D FFT) has recently been the most widely used experimental method for determining dispersion curves [22,23,24,25,26].

Recently, new modified methods have emerged that combine the advantages of several methods. The phase velocity filter described in [27] can extract waves of a certain phase velocity, regardless of the frequency. This technique exhibits reduced artefacts and is able to extract modes across the full bandwidth of the excitation.

The authors in [28] described 2D matrices that record the total response of a Lamb wave field in a limited surface measurement area. Using the discrete Fourier transform, a spectral estimate of the 2D matrix in the frequency-phase velocity domain is obtained. The variation of the phase velocity is mapped by moving the 2D matrix in the measurement area. The presented methodology has been investigated only for a specific plate, and its further development has not yet been analyzed.

The reliability of the measurement results of these methods is determined by the requirements that many measurements can be made at different distances and that the main result of the measurements is the phase velocity of the Lamb waves. Recently, new methodologies have been developed that allow the simultaneous measurement of both phase and group velocities of Lamb wave propagation.

The authors in [29] present an experimental methodology for the calculation of Lamb wave phases and group velocities. This technique uses conventional transducers to record two signals, spaced a few centimeters apart. The proposed method for calculating the dispersion curves has a number of limitations: highly dispersive modes are more challenging to evaluate, overlapping between wavepackets limits the method’s applicability, the size of the sample can be smaller at higher frequencies, etc.

A similar methodology is described in [30], in which an analytical cross-correlation method is used to determine the group delay and phase shift, which requires measurements at only two adjacent positions. The method was proposed for phase velocity estimation of fundamental Lamb wave modes. However, the presented method is not resistant to measurement noise and mode interference.

In [31], the phase and group velocities of the propagation of the Lamb wave are determined by measuring the propagation times. The propagation times were estimated using the zero-crossing technique. The phase velocity was estimated by setting zero-crossing instances, and these instances were fixed depending on the selected threshold. The essence of the method was to accurately measure the duration of each half-period of the signal at a selected distance. However, it has been found that the half-periods of different signals vary with the wave propagation in the plate and this affects the accuracy of the phase velocity determination. The measurement results are also influenced by the choice of threshold.

In this paper, we propose a new and simple algorithm for group and phase velocity determination based on signal filtering and zero-crossing estimation. The paper is organized as follows. In Section 2 the methodologies used for the determination of the group and phase velocities are described. In Section 3, the proposed method is analyzed using simulated signals, investigating the influences of the signal and filter parameters. The reliability of the proposed method using experimental signals from aluminum plate is evaluated in Section 4. Finally, Section 5 discusses the advantages and limitations of using the method, as well as outlining further research perspectives.

## 2. Methodology for Estimating Phase and Group Velocities

If a single Lamb wave mode is excited in an ideal plate of constant-thickness *d*, then a recorded signal *u*(*x*,*t*) with a propagation distance *x* can be described by the following equations [11,32]:(1)$$u\left(x,t\right)=\frac{1}{2\pi }\int_{-\infty }^{\infty }\mathrm{FT}\left(y\left(t\right)\right)e^{-jk\left(\omega \right)x}e^{j\omega t}d\omega =\frac{1}{2\pi }\int_{-\infty }^{\infty }\mathrm{FT}\left(y\left(t\right)\right)e^{-j\omega \frac{x}{c_{p}\left(\omega \right)}}e^{j\omega t}d\omega =\;\frac{1}{2\pi }\int_{-\infty }^{\infty }\mathrm{FT}\left(y\left(t\right)\right)e^{-j\int \frac{x}{c_{g}\left(\omega \right)}d\omega }e^{j\omega t}d\omega ,$$
where *y*(*t*) is the excitation signal, *t* is the time, FT is the Fourier transform, *k*(*ω*) is the wavenumber, *j* is the basic imaginary unit $j=\sqrt{-1}$, *ω* = 2π*f* is the angular frequency, *f* is the frequency, *c*_p_(*ω*) is the phase velocity and *c*_g_(*ω*) is the group velocity.

As we can see from Equation (1), the signal *u*(*x*,*t*), which has propagated the distance *x*, encodes information about the phase *c*_p_(*ω*) and group *c*_g_(*ω*) velocities. Therefore, the aim of this work was to extract this information from these signals. A filter packet and a zero-crossing method were used for this.

To explain the algorithm used for calculating the Lamb wave velocities, let us take a typical B-scan image of signal propagation in a *d* = 1 mm thick aluminum plate at an excitation frequency of *f*_ex_ = 300 kHz (Figure 1a). As the excitation signal *y*(*t*), a three-period harmonic signal with a Gaussian envelope was used. The signals propagated at the distances *x*_1_ = 40 mm and *x*_2_ = 160 mm, respectively, are shown in Figure 1b. The envelopes of these signals are also presented, the peaks of which determine the location of the maximum energy concentration of the signals:(2)$$e\left(x,t\right)=\mathrm{HT}\left[u\left(x,t\right)\right],$$
where HT is the Hilbert transform.

With this assumption in mind, the idea was to capture the signal’s zero-crossing instances closest to the signal’s envelopes peaks, and then to use these time instances to determine the phase and group velocities.

To implement the idea, a filter packet of selected parameters was used to filter the signals. Then, filtered signals *s_i_*(*x*,*t*) at the propagated distance *x* can be described:(3)$$s_{i}\left(x,t\right)=\frac{1}{2\pi }\int_{-\infty }^{\infty }\mathrm{FT}\left(y\left(t\right)\right)B_{i}\left(\omega \right)e^{-j\omega \frac{x}{c_{p}\left(\omega \right)}}e^{j\omega t}d\omega ,$$
where *B_i_*(*ω*) represents the frequency response of *i*-th bandpass filter, *I* = 1, 2, …, N, where N is the total number of filters.

To reduce the reconstruction error due to the wave decomposition process, we allow each filter to have a Gaussian magnitude function [33]:(4)$$B_{i}\left(f\right)=e^{-2.77\cdot {\left(\frac{f-f_{L}-\left(i-1\right)df}{\Delta B}\right)}^{2}},$$
where *f*_L_ is the lower frequency limit, $df=\frac{f_{H}-f_{L}}{N-1}$ is the step in the frequency domain between the central frequencies of two neighbouring filters, *f*_H_ is the upper frequency limit and Δ*B* is the filter bandwidth. The upper and lower frequency limits are selected according to the frequency response of the test signal *u*(*x*,*t*) at the −6 dB level (level of 0.5). Thus, the signal bandwidth is $\Delta f=f_{H}-f_{L}$.

When filtering the dispersed signals using the filtering algorithm [20], it was observed that the signals filtered using different filters *s_i_*(*x*,*t*) have zero-crossing instances on the time axis, which are concentrated in the signal envelope peak environment (Figure 2a,b). The signal envelope peak environment is treated as the time interval corresponding to the envelope at the −6 dB level.

Therefore, in our subsequent studies, only the zero-crossing instances in the signal envelope peak environment were selected. The analyzed zero-crossing instances to both sides of the maximum were selected according to the following equations:(5)$$t_{\mathrm{emax}}=\arg\left\{\max\left(e\left(x,t\right)\right)\right\},$$
(6)$$t_{ik}^{0}=\arg\left\{s_{i}\left(x,t\right)≅0\right\},\;\mathrm{if}\;t_{\mathrm{emax}}-\frac{5}{2f_{i}}<t<t_{\mathrm{emax}}+\frac{5}{2f_{i}},$$
where $t_{ik}^{0}$ is the zero-crossing instances of *i*-th filter, *k* = 1, 2, …, K is the number of zero-crossing instances; K is the total number of zero-crossing instances and *f_i_* is the central frequency of the *i*-th filter.

Concentrated zero-crossing instances on the time axis were determined according to the minimum time difference between them:(7)$$t_{iM}^{0}=\arg\left\{\min\left(\sum_{i=1}^{N-1}\left(\underset{1<k<K}{\min}\left(t_{ik}^{0}-t_{\left(i+1\right)k}^{0}\right)\right)\right)\right\},$$
where $t_{iM}^{0}$ is concentrated zero-crossing instances and M is the number of zero-crossing instance in *i*-th filter.

As a result of these calculations, we obtained one zero-crossing instance $t_{iM}^{0}$ for each filtered signal *s_i_*(*x*,*t*). In the next stage we calculated these zero-crossing instances $t_{iM}^{0}\left(x\right)$ for each value of distance *x*. The obtained results are presented in a B-scan image, together with the investigated signals, in Figure 3a.

As can be seen from the obtained results, the dependence of the zero-crossing instances on the distance in the narrow ranges was linear. However, at some distances, there were jumps in the line of the zero-crossing instances (Figure 3b). This phenomenon has been noted earlier in [31]. This was based on the fact that the phase and group velocities differ, and the half-periods of the signal “move” inside the signal envelope as the distance changes. Two such jumps can be described by means of a set of four points (*x_i_*_(1–4)_,*t_i_*_(1–4)_) (Figure 3b). During this study, it was observed that line 1 was formed between the two jumps (*x_i_*_2_÷*x_i_*_3_), which consisted of zero-crossing instances of equal phase of the filtered signals. From the zero-crossing instances of the same phase, the phase velocity of the propagation of the Lamb waves can be calculated as follows.
(8)$$c_{pi}\left(x_{i2}÷x_{i3}\right)=\frac{x_{i3}-x_{i2}}{t_{i3}^{0}-t_{i2}^{0}}$$

Meanwhile, the nature of the signal envelope propagation was described by line 2, which corresponds to the group velocity of the Lamb wave propagation:(9)$$c_{gi}\left(x_{i2}÷x_{i4}\right)=\frac{x_{i4}-x_{i2}}{t_{i4}^{0}-t_{i2}^{0}},$$

Both the phase *c*_p*i*_(*x_i_*_2_ ÷ *x_i_*_3_) and the group *c*_g*i*_(*x_i_*_2_ ÷ *x_i_*_4_) velocities of each filtered signal *s_i_*(*x*,*t*) for such a distance segment between two jumps (*x_i_*_1_ ÷ *x_i_*_3_) can be calculated.

## 3. Investigation of the Proposed Method Using Simulated Signals

Validation of the proposed method was performed using simulated signals. The general verification algorithm of the proposed method is presented in Figure 4. The objective of the verification was to calculate the mean relative error between the values of the dispersion curves of the simulated groups and phase velocities in the selected frequency range and the values calculated by the proposed method in this frequency range.

A *d* = 1 mm thick aluminum 7075-T6 plate with the following parameters was selected for modeling the propagation of Lamb wave A_0_ mode signals: Young modulus *E* = 71.7 GPa, Poisson’s ratio *ν* = 0.33 and density *ρ* = 2710 kg/m^3^. According to these parameters, the phase and group velocity dispersion curves of Lamb wave A_0_ mode propagation in such a plate were calculated by means of the one-dimensional SAFE method [4] (Figure 5).

Based on the phase velocity curve of the Lamb wave A_0_ mode propagation in the aluminum plate (Figure 5), the B-scan images of different wave excitation frequencies *f*_ex_ (according to Equation (1)) were formed: 100, 300 and 700 kHz. These frequencies were selected at different locations for the variations of the phase and group velocities.

In the modelling, a three-period harmonic signal with a Gaussian envelope was used as the excitation signal *y*(*t*). The waveform of the transmitted signal was calculated at a distance of 200 mm with a step of d*x* = 0.1 mm. The B-scan images of the simulated Lamb wave A_0_ mode, propagated in *d* = 1 mm thick aluminum plate at different wave excitation frequencies *f*_ex_, are presented in Figure 6a–c.

Filter packets of selected parameters were used to filter the signals of the simulated B-scan images. The lower *f*_L_ and upper *f*_H_ frequency limits were selected according to the frequency responses (FR) of the simulated signals at the −6 dB level (level of 0.5) (Figure 7a). Different signal bandwidths $\Delta f=f_{H}-f_{L}$ were obtained for different excitation frequencies *f*_ex_: Δ*f* = 53.3 kHz for *f*_ex_ = 100 kHz, Δ*f* = 160 kHz for *f*_ex_ = 300 kHz and Δ*f* = 376.7 kHz for *f*_ex_ = 700 kHz. The center frequencies of the filters were selected according to the frequency characteristics of the signals. The conditions were chosen so that the center frequency of the lowest frequency filter corresponded to the frequency *f*_L_, the center frequency of the highest frequency filter corresponded to the frequency *f*_H_, and the center frequency of the middle filter corresponded to the maximum frequency of the signal frequency response (Figure 7a). In this way, by selecting the number of filters in the filter packet, the number of filters was three, five, seven, …, The next parameter selected was the individual filter bandwidth Δ*B*. The ratio of the bandwidth of the signal frequency response to the bandwidth of the individual filter R = Δ*f*/Δ*B* was used to select this parameter. The bandwidths of all individual filters were selected to be the same. Figure 7a shows an example of how five filters with bandwidths R = 2.5 were selected for a 300 kHz signal. The frequency response of the signal and the total frequency response of the filters are shown next to them (Figure 7b). The values of phase and group velocities calculated for the given parameters by means of the above-described algorithm are shown in Figure 7c. It should be noted that the phase velocity values were obtained as much as the filters were used. Meanwhile, only one value corresponding to the center frequency of the signal was obtained for the group velocity.

The measurements presented in Figure 7c were performed for all cases of excitation frequency *f*_ex_ signals. After theoretical modeling, it was observed that the main parameters influencing the accuracy of phase and group velocity calculations were the number of filters (N) and the bandwidths (Δ*B*) of individual filters.

In order to evaluate the suitability of the proposed method, the mean relative error $\delta_{cp\left(g\right)}$ was used for the comparison of the results obtained using the proposed method and using the SAFE method:(10)$$\delta_{cp\left(g\right)}=100\%\cdot \frac{1}{N}\sum_{i=1}^{N}\frac{\left|c_{ip\left(g\right)}-c_{p\left(g\right)}^{\mathrm{SAFE}}\right|}{c_{p\left(g\right)}^{\mathrm{SAFE}}},$$
where *c_i_*_p(g)_ are the values of the phase (group) velocity calculated using the proposed algorithm, $c_{p\left(g\right)}^{\mathrm{SAFE}}$ are the values of the phase (group) velocity calculated using the SAFE method at the same frequencies and N is total number of the filters. The calculation results of the mean relative error $\delta_{cp\left(g\right)}$ are presented in Table 1.

The analysis of the results relating to the mean relative error $\delta_{cp\left(g\right)}$ in Table 1 provided interesting information. The phase velocity calculation error was the lowest for all excitation frequencies *f*_ex_ when seven filters with bandwidth ratio R = 3 were used. In addition, the values of the mean relative error differed quite slightly (0.57% for *f*_ex_ = 100 kHz, 0.67% for *f*_ex_ = 300 kHz, 0.53% for *f*_ex_ = 700 kHz). This difference can be explained by the varying nature of the dispersion and the different bandwidths at the analyzed frequencies. The obtained results suggest that an even larger number of filters is required. However, there was no need to further increase the number of filters as uncertainties in the determination of zero-crossing concentrations began to emerge (Equation (7)).

Different trends prevailed for group velocity calculations. Since the group velocity was calculated only in the case of the middle filter, the number of filters does not affect its value. However, the bandwidth of the filter affects the mean relative error. The smallest error (1.75% for *f*_ex_ = 100 kHz, 0.49% for *f*_ex_ = 300 kHz, 0.42% for *f*_ex_ = 700 kHz) was obtained when using the filter bandwidth of the signal bandwidth (R = 1). A decrease in error was also observed at a narrower filter band (R = 3), but only at higher frequencies (0.74% for *f*_ex_ = 300 kHz, 0.53% for *f*_ex_ = 700 kHz).

Theoretical modeling has shown that the proposed method enables the calculation of group and phase velocities with a mean relative error of less than 0.7% using simulated signals. In the case of experimental studies, these errors can change significantly. Therefore, in the subsequent stage, experimental studies were performed and the mean relative errors of group and phase velocity calculations were estimated.

## 4. The Reliability of the Proposed Method Using Experimental Signals

Quantitative evaluation of the proposed method for measuring the phase and group velocity of Lamb waves was performed using A_0_ mode propagation experimental data sets in a *d* = 2 mm thick aluminum plate (1.2 × 1.2 m^2^). Figure 8 presents a structural scheme of the experimental equipment used in the study. The experiments were performed using the ultrasonic measuring system “Ultralab” and an axis driver developed at the Ultrasound Research Institute of Kaunas University of Technology. The position of the ultrasonic moving receiver was changed with a Standa 8MTF-75LS05 scanner (Standa Ltd., Vilnius, Lithuania).

The A_0_ mode of Lamb waves was generated in the aluminum plate by contact transducers, excited at a resonant frequency of *f*_ex_ = 160 kHz. Contact-point-type transducers with a hemispherical plastic tip were used. The excitation signal was a three-period burst with the Gaussian envelope. A B-scan image was formed when the receiving transducer moved at a distance of 60–260 mm with 0.1 mm steps (Figure 9a). Figure 9b shows the recorded A_0_ mode signals at different distances, *x*_1_ = 80 mm and *x*_2_ = 220 mm. The amplitudes of all B-scan image signals *u*(*x*,*t*) were normalized to the maximum amplitude of the first received signal *u*(*x*_0_,*t*) (*x*_0_ = 60 mm). The next figure (Figure 9c) shows the amplitude frequency responses (FR) of the displayed signals *u*(*x*_1_,*t*) and *u*(*x*_2_,*t*).

The required frequency band parameters of the filter packet were selected according to the width of the amplitude frequency response of the determined signals. The signal bandwidth (Δ*f* = 46.1 kHz) was calculated based on the determined lower *f*_L_ = 138.1 kHz and upper *f*_H_ = 184.2 frequency values. Based on the theoretical research, seven filters (*n* = 7) with the frequency bandwidth ratio R = 3 were selected. The resonant frequency of the central filter was *f*_4_ = 161.15 kHz and the distance between the filters was *df* = 7.68 kHz.

Figure 10a presents a Lamb wave A_0_ mode experimental B-scan image (colour coded) with the calculated zero-crossing instances (line). In jump-limited intervals, phase and group velocities were calculated with the coordinates of the distance of each interval as the center of the interval.

The results of the calculations are presented in Figure 10b (dots), where different ranges of phase and group velocity changes are shown separately. Seven phase velocity curves (both filters are used) and one group velocity curve were formed. After calculating the averages of the phase and group velocity changes, the mean absolute errors $\Delta_{p\left(g\right)}^{i}$ and the mean relative errors $\delta_{p\left(g\right)}^{i}$ of the following velocities from the average were estimated for each filter separately:(11)$${\bar{c}}_{p\left(g\right)}^{i}=\frac{1}{Q}\sum_{q=1}^{Q}c_{p\left(g\right)q}^{i},\;\Delta_{p\left(g\right)}^{i}=\frac{1}{Q}\sum_{q=1}^{Q}\left|c_{p\left(g\right)q}^{i}-{\bar{c}}_{p\left(g\right)}^{i}\right|,\;\delta_{p\left(g\right)}^{i}=100\%\cdot \frac{1}{Q}\sum_{q=1}^{Q}\frac{\left|c_{p\left(g\right)q}^{i}-{\bar{c}}_{p\left(g\right)}^{i}\right|}{{\bar{c}}_{p\left(g\right)}^{i}},$$
where ${\bar{c}}_{p\left(g\right)}^{i}$ is the phase (or group) velocity average, *q* is number of the value of velocity and *q* = 1, 2, …, Q, Q is the total number of the velocity values (jump-limited intervals). The calculation results are presented in Table 2.

The results of the measurements and calculations show that the values of the phase and group velocities in the isotropic material (aluminum) were measured stably and with a small scattering relative to the mean (<0.8%).

A comparison with other methods should be performed to verify the proposed method. For this purpose, a widely used 2D-FFT method [22] was chosen to evaluate the results of the phase velocity calculations. The result of the 2D-FFT method is a two-dimensional image, generated using the total B-scan image data (Figure 11, colored). Therefore, the average values of the phase velocity over the measured distance were used for data comparison. The obtained comparison results are presented in Figure 11. In summary, the results of the phase velocity calculation obtained using the proposed method and the 2D FFT method were consistent. The mean absolute error was Δ_p_ = 4.7 m/s and the mean relative error was *δ*_p_ = 0.3%.

The correlation method described in [34] was used to calculate the group velocity. The delay time Δ*t* between the two envelopes *e*(*x*_1_,*t*) and *e*(*x*_2_,*t*) of the signals *u*(*x*_1_,*t*) and *u*(*x*_2_,*t*) was estimated using cross-correlation:(12)$$\Delta t=\arg\;\max\left\{corr\left[e\left(x_{1},t\right),e\left(x_{2},t\right)\right]\right\}.$$

Then the group velocity was estimated:(13)$$c_{g}=\frac{x_{2}-x_{1}}{\Delta t}.$$

The group velocity was calculated 10 times by taking different distances (Figure 9a) between the signals and averaging the values obtained. The value of the average group velocity was $\bar{c_{g}}$ = 2652.8 m/s. Comparing this value with the experimentally measured value (Table 2) gave the mean absolute error Δ_g_ = 15.3 m/s and the mean relative error *δ*_g_ = 0.6%.

## 5. Discussion and Conclusions

This paper presents a new and simple experimental algorithm for Lamb wave A_0_ mode group and phase velocity measurements. The new tool based on a filter packet and a zero-crossing method were used to process the signals. The proposed method captures the zero-crossing time instances of the signal closest to the signal envelope maximum and simultaneously determines the phase and group velocities using these time instances. The reliability of the proposed method was evaluated using simulated and experimental signals propagating in an aluminum plate. Theoretical modeling in a 1 mm thick aluminum 7075-T6 plate showed that the proposed method enabled the calculation of group and phase velocities with a mean relative error of less than 0.7% using simulated signals. An accuracy of 0.8% was observed during the experimental measurements in a 2 mm thick aluminum plate. The obtained results showed that the proposed method of group and phase velocity estimation enables researchers to calculate the segments of the A_0_ mode dispersion curve of isotropic materials.

However, this method has some limitations and unexplored potential applications. The overlapping of different modes limits the application of the method, as it distorts the phase of the analyzed mode. The determination of the phase and group velocities requires a certain scanning distance, which is conditioned by a fixed jump in the propagation of the A_0_ mode. The duration of these jumps and the influence of optional parameters on this duration need to be examined in further studies. Another unanswered question is how the occurrence of a defect in the scan trajectory or a change in the plate thickness affects the determination of the phase and group velocity. The application of this method to complex composite plates is also relevant. The application of this methodology to a non-dispersive mode (S_0_ mode) has not been investigated either. All these issues will be studied in further work.

Despite the listed shortcomings, further investigation of this method is promising, as this method can simultaneously measure the group and phase velocities and can complete this process on-line. In our further research, we envisage the possibility of applying this method in studies of the spatial distribution of phase and group velocities. Further research would include the application of this method to address SHM problems when group and phase velocities are used directly as qualitative indicators.

## Figures and Tables

**Figure 1 materials-15-02976-f001:**
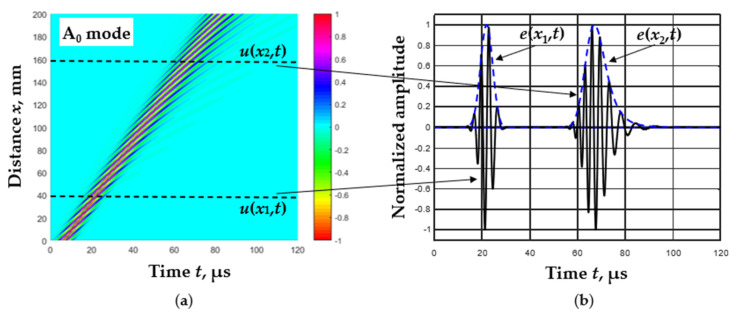
B-scan image of the simulated Lamb wave A_0_ mode in *d* = 1 mm thickness aluminum plate at *f*_ex_ = 300 kHz excitation frequency (**a**) and the waveforms of received signals at the distances *x*_1_ = 40 mm and *x*_2_ = 160 mm with its envelopes (**b**).

**Figure 2 materials-15-02976-f002:**
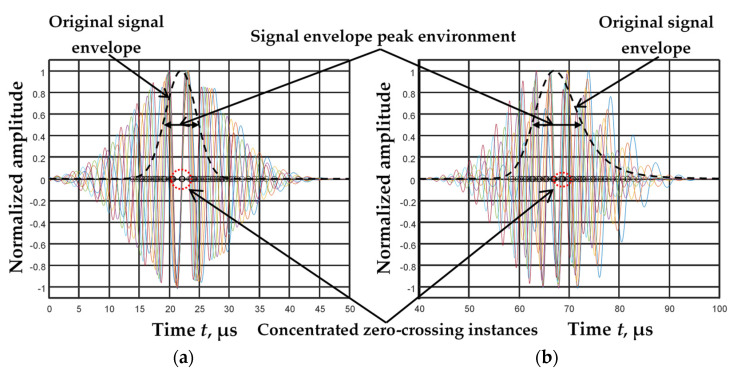
Signals filtered by means of five filters (color curves) *s_i_*(*x*,*t*) at the distances *x*_1_ = 40 mm (**a**) and *x*_2_ = 160 mm (**b**) with the envelopes of the original signals *u*(*x*,*t*).

**Figure 3 materials-15-02976-f003:**
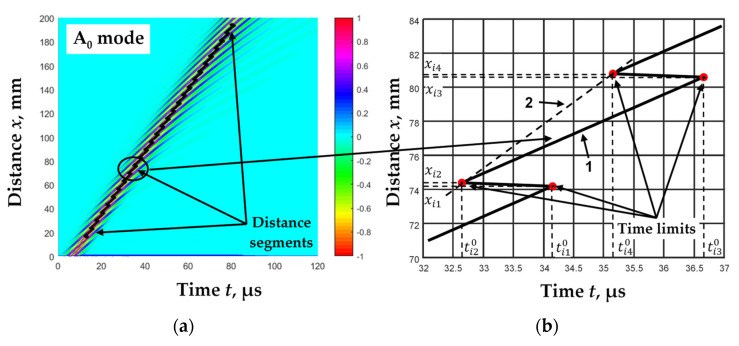
Calculated zero-crossing instances (line) with simulated Lamb wave A_0_ mode signals (color-coded) (**a**) and zero-crossing instances of the *i*-th filter in the narrow range (**b**).

**Figure 4 materials-15-02976-f004:**
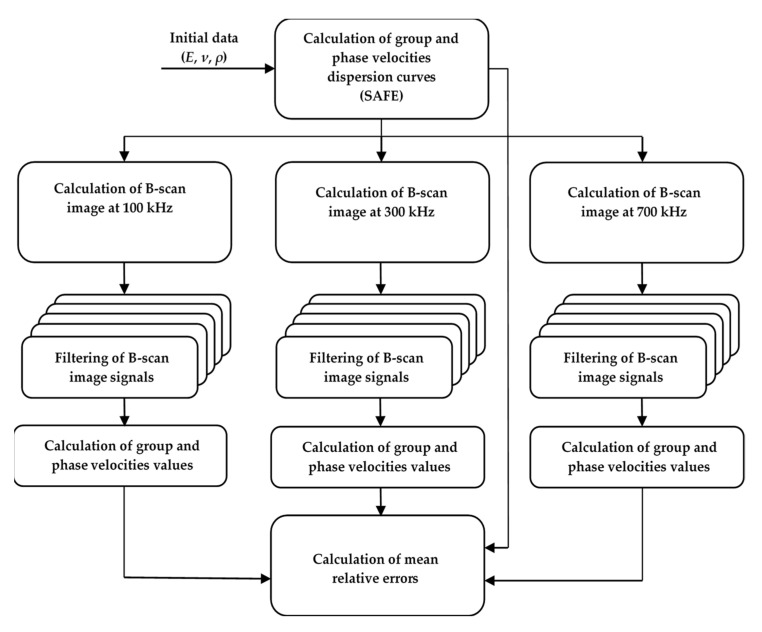
General verification algorithm of the Lamb wave A_0_ mode group and phase velocity calculation method.

**Figure 5 materials-15-02976-f005:**
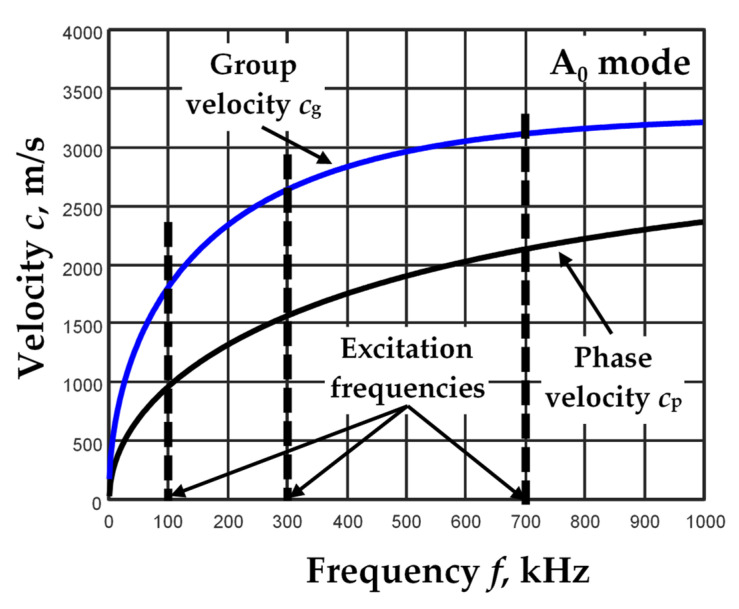
The Lamb wave A_0_ mode group and phase velocity dispersion curves in 1 mm thick aluminum plate, calculated using the one-dimensional SAFE method.

**Figure 6 materials-15-02976-f006:**
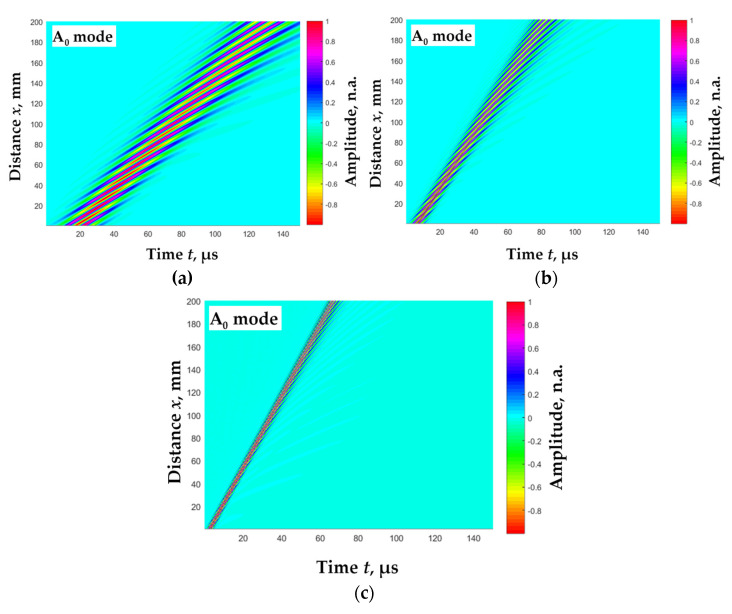
The B-scan images of the simulated Lamb wave A_0_ mode propagated in *d* = 1 mm thick aluminum plate at different wave excitation frequencies *f*_ex_: (**a**) 100 kHz, (**b**) 300 kHz, (**c**) 700 kHz.

**Figure 7 materials-15-02976-f007:**
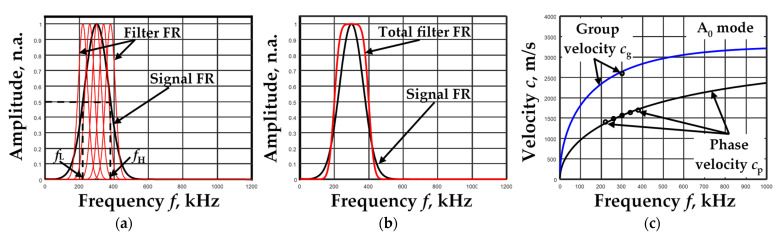
The frequency responses (FR) of the *f*_ex_ = 300 kHz signal and *n* = 5 filters when using filter bandwidth R = 2.5 (**a**), total frequency response of the filters (**b**), values of phase and group velocities calculated for the proposed algorithm (dots) and dispersion curves calculated using the SAFE method (lines) (**c**).

**Figure 8 materials-15-02976-f008:**
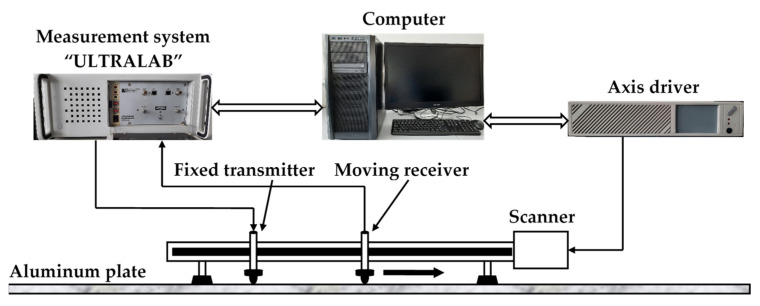
The structural scheme of Lamb wave A_0_ mode signal generation and recording in an aluminum plate.

**Figure 9 materials-15-02976-f009:**
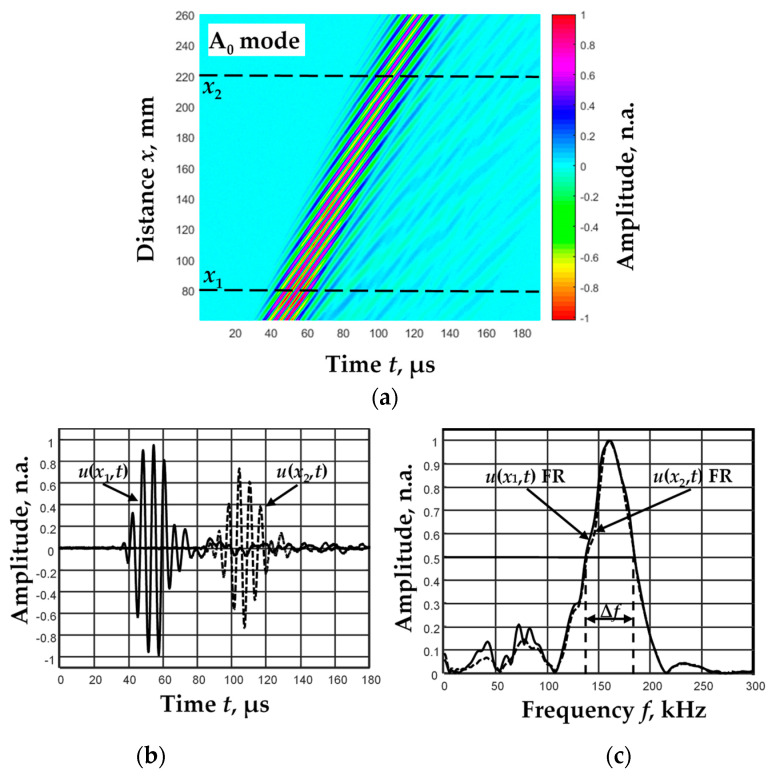
B-scan image of the A_0_ mode propagating through an aluminum plate at a test distance of 60 to 260 mm from the excitation point (**a**), the signals *u*(*x*_1_,*t*) and *u*(*x*_2_,*t*) at the distances *x*_1_ = 80 mm and *x*_2_ = 220 mm (**b**) and the frequency responses (FR) of these signals (**c**).

**Figure 10 materials-15-02976-f010:**
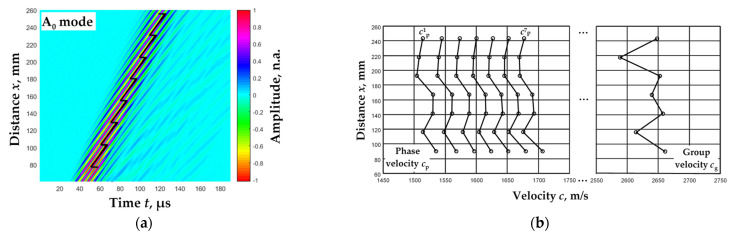
Lamb wave A_0_ mode B-scan image (color-coded) with calculated zero-crossing instances (line) (**a**) and values of phase and group velocities calculated for the proposed algorithm (dots) (**b**).

**Figure 11 materials-15-02976-f011:**
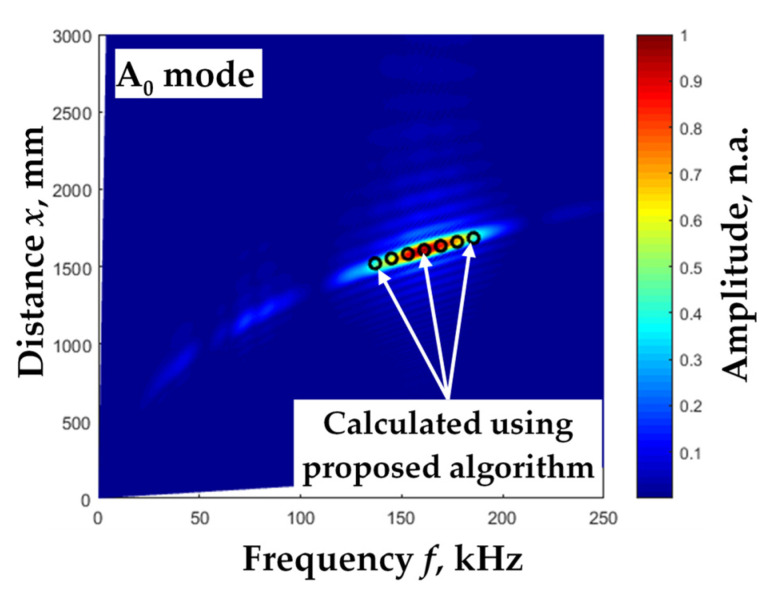
The experimentally calculated phase velocity dispersion curve values of the Lamb wave A0 mode using the proposed algorithm (dots) and the calculated phase velocity dispersion curve using 2D FFT method (color).

**Table 1 materials-15-02976-t001:** The calculations of the mean relative error of the phase and group velocities using the proposed method.

Excitation Frequency *f*_ex_, kHz	The Ratio of the Bandwidth of the Signal Spectrum to the Bandwidth of the Filters R = Δ*f*/Δ*B*	Number of Filters N					
		3		5		7	
		Mean Relative Error *δ_c_*, %					
		*δ_c_* _p_	*δ_c_* _g_	*δ_c_* _p_	*δ_c_* _g_	*δ_c_* _p_	*δ_c_* _g_
**100**	1	4.35	1.75	4.03	1.75	3.63	1.75
	1.5	2.60	1.92	2.48	1.92	2.16	1.92
	2	1.60	2.03	1.60	2.03	1.33	2.03
	2.5	1.03	2.09	1.09	2.09	0.86	2.09
	3	0.68	2.89	0.79	2.89	0.57	2.89
**300**	1	4.05	0.49	3.56	0.49	3.36	0.49
	1.5	2.50	1.61	2.20	1.61	2.07	1.61
	2	1.63	1.68	1.42	1.68	1.34	1.68
	2.5	1.12	1.72	0.98	1.72	0.92	1.72
	3	0.81	0.74	0.71	0.74	0.67	0.74
**700**	1	3.19	0.42	2.80	0.42	2.65	0.42
	1.5	1.99	0.47	1.74	0.47	1.64	0.47
	2	1.29	0.50	1.13	0.50	1.06	0.50
	2.5	0.89	0.52	0.78	0.52	0.73	0.52
	3	0.64	0.53	0.56	0.53	0.53	0.53

**Table 2 materials-15-02976-t002:** The calculations of the velocity averages, the mean absolute errors and the mean relative errors of the phase and group velocities.

	Velocity							
	Phase Velocity *c*_p_							Group Velocity *c*_g_
	Filter Number *i* (Corresponding Frequency, kHz)							
	1 (138.1)	2 (145.8)	3 (153.5)	4 (161.1)	5 (168.8)	6 (176.5)	7 (184.2)	
**Velocity** **Average, m/s**	1519.3	1551.2	1580.5	1606.9	1632.9	1658.6	1683.3	2637.5
**Mean Absolute Error, m/s**	10.7	10.2	9.4	9.4	10.2	11.1	11.5	20.6
**Mean Relative Error, %**	0.7	0.7	0.6	0.6	0.6	0.7	0.7	0.8

## Data Availability

Not applicable.

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
