# Peer review of "Experimental Method for Simultaneous Determination of the Lamb Wave A0 Modes Group and Phase Velocities"

_materials, 2022, doi:10.3390/ma15092976_

Round 1
Reviewer 1 Report
No comments on the quality of the research - the method is very well described and results seem solid.
However, I'd strongly suggest having your manuscript proofread by a native speaker to drastically increase its legibility.
Author Response
I would like to thank Reviewer 1 for taking the time and reviewing this article.
Reviewer 2 Report
- The authors propose a new and simple experimental algorithm for A0 mode group and phase velocity measurement based on signal filtering and zero-crossing estimating. The innovative concept is to capture the zero crossing instances of the signals closest to the signal envelope peaks and use these time instances to deter- mine the phase and group velocities.
- Their analysis results show that the proposed method enables the calculation of the A0 mode group and phase velocities with a mean relative error of less than 0.7 %. The accuracy of 0.8 % was determined during the experimental measurements.
- In the figure 5, the Lamb wave A0 mode group and phase velocities dispersion curves in 1 mm thickness aluminium plate calculated using one-dimensional SAFE method, should be demonstrated in detail.
- Please revise and enhance the English writing to improve the manuscript’s readability.
Author Response
My answers are presented in the attached document.

Reviewer 3 Report
The paper have proposed a new algorithm for group and phase velocity measure based on signal filtering and zero-crossing estimation. Although for application, the method have some limitations overlapping with different mode, the authors described the limitation in the discussion.
Author Response
I would like to thank Reviewer 3 for taking the time and reviewing this article.
Reviewer 4 Report
This manuscript proposed a method to determine the phase and group velocities of A0 mode Lamb waves. The feasibility of proposed method has been verified through simulation and experiments. The followings parts can be revised to improve the quality of the paper.
Major:
- How does the jump of Line 1 in Figure 3 related to phase velocity? How does Line 2 related to group velocity? Please add detailed explanations in the manuscript.
- The authors specially emphasized that the proposed method is suitable for A0 mode Lamb waves. Is this method suitable for S0 mode? It is recommended to add the discussion in Section 4.
- For the signals obtained from simulation and experiments, is it able to use previous methods that were introduced in the introduction to calculate phase and group velocities? It’s better add a comparison of the accuracy to shown the advantages of the newly proposed method.
- Is there any frequency limit for the proposed method? Is it able for the method to be applied to get velocities at several kilo-Hertz or several mega-Hertz?
Minor:
- In Line 52 and 53 of Page 2, STFFT is the abbreviation of short time fast Fourier transform, while in the text, the authors wrote short time Fourier transform. It is recommended to either change STFFT to STFT, or change short time Fourier transform to short time fast Fourier transform.
- In Line 135 of Page 3, it should be x2=160mm.
- Legends or descriptions should be added in Figure 2.
Author Response

(The authors gave the same response as above.)

Reviewer 5 Report
This manuscript proposes the method for the determination of phase and group velocity from Lamb wave signals. This could be useful for non-destructive testing applications. The novelty compared to [31] should be demonstrated and what is the efficiency of the method: can the information obtained from two signals or signal set is needed. Major revision.
Comments:
Title: measurement is not an appropriate term, rather a determination, direct measurement is not possible, should be rephrased also elsewhere.
Introduction: some paragraphs contain only one, or two sentences, better presentation is needed.
Line 28: Their main disadvantage, and thus their advantage, is their dispersive nature - the phase and group velocities vary with frequency. Not a clear sentence.
Can you explain in the first paragraph why it is difficult to determine phase and group velocity from a dispersive wave packet?
Can you bring out the novelty compared to [31] and highlight the main advantage compared to other techniques?
Methodology:
Fig.1: how many cycles and how is windowed the excited signal? Information is on page 7 but should be shown also here.
Not clear what do different colors in Fig. 2 mean? For clarity separate one filtered result and explain it.
Not clear what does "signal envelope peak environment" mean?
The first signal was measured at 40 mm. Not clear why the range from 70 to 84 mm in Fig.3 is shown. Is it somehow extrapolated to this region from 40 mm? Can you explain how the information from Fig.2 is used to get Fig.3 b.
Do you need only two signals or a measurement set to determine the velocities? If you need a measurement set then what is the advantage compared to 2D FFT method? How important is the measurement step (mm) in the results?
Give the physical explanation to (8) and (9).
In Table 1, define delta_cp and delta_cg in the caption.
Author Response

(The authors gave the same response as above.)

Reviewer 6 Report
To the reviewer's opinion, the paper is very rigorous and could be published in the actual state.Author Response
I would like to thank Reviewer 6 for taking the time and reviewing this article.
Round 2
Reviewer 2 Report
no further comment.
Reviewer 4 Report
The authors replied to all my comments. I recommend publication in present form.
Reviewer 5 Report
Thanks for the answers, I believe that the manuscript now meets the publishing requirements. I support its publishing.